# Complementary Analysis for DGA Based on Duval Methods and Furan Compounds Using Artificial Neural Networks

**Ancuța-Mihaela Aciu** [1], **Claudiu-Ionel Nicola** [1,2,*], **Marcel Nicola** [1,*] **and Maria-Cristina Nițu** [1,3]

[1] Research and Development Department, National Institute for Research, Development and Testing in Electrical Engineering-ICMET Craiova, 200746 Craiova, Romania; ancutu13@yahoo.com (A.-M.A.); cristinamarianitu@yahoo.com (M.-C.N.)

[2] Department of Automatic Control and Electronics, University of Craiova, 200585 Craiova, Romania

[3] Department Electrical Engineering, Energetic and Aeronautics, University of Craiova, 200585 Craiova, Romania

*   Correspondence: claudiu@automation.ucv.ro (C.-I.N.); marcel_nicola@yahoo.com (M.N.)

**Abstract:** Power transformers play an important role in electrical systems; being considered the core of electric power transmissions and distribution networks, the owners and users of these assets are increasingly concerned with adopting reliable, automated, and non-invasive techniques to monitor and diagnose their operating conditions. Thus, monitoring the conditions of power transformers has evolved, in the sense that a complete characterization of the conditions of oil–paper insulation can be achieved through dissolved gas analysis (DGA) and furan compounds analysis, since these analyses provide a lot of information about the phenomena that occur in power transformers. The Duval triangles and pentagons methods can be used with a high percentage of correct predictions compared to the known classical methods (key gases, International Electrotechnical Commission (IEC), Rogers, Doernenburg ratios), because, in addition to the six types of basic faults, they also identify four sub-types of thermal faults that provide important additional information for the appropriate corrective actions to be applied to the transformers. A new approach is presented based on the complementarity between the analysis of the gases dissolved in the transformer oil and the analysis of furan compounds, for the identification of the different faults, especially when there are multiple faults, by extending the diagnosis of the operating conditions of the power transformers, in terms of paper degradation. The implemented software system based on artificial neural networks was tested and validated in practice, with good results.

**Keywords:** power transformer; insulation; dissolved gas analysis; furan compounds; radial basis function neural network; feed forward neural network

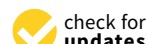



## 1. Introduction

Power transformers are key pieces of equipment in the electric power transmission and distribution systems, and their reliability influences the safety of electric power supply networks. Although they are reliable pieces of equipment, it is difficult to avoid damaging them. In addition to catastrophic damage to the power supply system, the faults in transformers can also cause substantial financial losses for both the owner and the consumers served by it. For this reason, it is important to identify, at an early stage, the possible faults in transformers so that, based on an appropriate diagnostic procedure, an efficient and rational decision is taken in advance on the appropriate corrective actions to be applied to the transformer [1–4].

An important and effective tool for the early-stage fault diagnosis in oil-immersed power transformers is dissolved gas analysis (DGA), which can identify the degradation of the solid insulation and oil [5–10]. The diagnosis of the involvement of solid insulation and its possible carbonization, resulting from a method of dissolved gas analysis, can be

confirmed by careful use of carbon oxides, the $CO_2/CO$ ratio, and the analysis of furan compounds [11–15].

Lately, in order to correct and render, more effectively, the methods used for identifying faults in oil-immersed power transformers that rely on the DGA, researchers around the world have tried to apply, to these methods, various techniques based on artificial intelligence, such as the fuzzy logic-based method proposed in [16], as a new solution for determining the fault condition using the combination of gas level, gas rate, and DGA interpretation through the Duval pentagon method.

One approach carrying out the classification of power transformer faults, based on the combination of intelligent methods, such as the hypersphere multi-class support vector machine (HMSVM), the hybrid immune a logarithm (HIA), and the kernel extreme learning machine (KELM), is shown in [17]. The optimization of the parameters of the HMSVM-type method, starting from the training stage, is achieved by means of particle swarm optimization (PSO). The fusion of these methods is achieved by the Dempster–Shafer (DS) evidence theory in order to increase the accuracy of the results.

Starting from the problem of insufficient and imbalanced datasets, one method that carries out the diagnosis of power transformers and overcomes these issues is the twin support vector machines (TWSVMs) method, proposed in [18]. The parameters of this intelligent classification method are optimized by using the chemical reaction optimization (CRO) type algorithm, and the efficiency and accuracy are increased by means of restricted Boltzmann machines (RBMs) from the moment of data preprocessing.

Moreover, to avoid a series of drawbacks regarding the traditional methods of DGA, an approach based on an improved algorithm of the grey wolf type, in order to get better results in case of fault classification by the least square support vector machine (LSSVM), is proposed in [19].

Based on the DGA and the construction of a global health index of power transformers, intelligent algorithms based on the improved differential evolution optimization algorithm, which substantially improve the precision and accuracy in determining the faults, are used in [20,21].

A complementarity for DGA, which is carried out through the partial discharge study by using the correlation analysis and extraction of the main characteristic parameters is shown in [22].

Moreover, a complex image of the fault condition of power transformer is presented in [23] and consists of determining the winding hot spot temperature combined with the DGA.

Although the intelligent methods and theories presented above have yielded good results on the accuracy of the power transformer fault diagnosing methods based on the DGA, there are also some shortcomings in the parameter optimization, the selection of the set of characteristics and the data preprocessing methods, which narrow the practical application of the intelligent systems [1,17,18].

The novelty presented in this article consists in the complementarity between the analysis of the gases dissolved in the oil of power transformers and the analysis of furan compounds, for the identification of the different faults, especially when there are multiple faults, by extending the diagnosis of the operating condition of power transformers in terms of paper degradation. This automatic and non-invasive diagnostic is based on the Duval triangles and pentagons analysis methods. The software implementation of the diagnosis is based on using artificial neural networks, such as Radial Basis Function Neural Network (RBFNN) and Feed Forward Neural Network (FFNN), due to their facilities, such as: learning, robustness of the algorithms used, and good results on accuracy and precision.

This paper is a continuation of the previous research [24–26] related to the determination of power transformer faults based on the fuzzy logic implementation of the DGA and the analysis of furan compounds.

The rest of the paper is organized as follows: Section 2 presents the methods for the analysis of dissolved gases and furan compounds. Section 3 presents the system developed

to determine the faults in oil-immersed power transformers, based on neural networks. Section 4 presents the validation of the proposed system, and the conclusions and future approaches are presented in Section 5.

## 2. Materials and Methods for Faults Detection of the Power Transformers

The interpretation of the DGA in oil-filled transformers is one of the most important procedures for the determination of the types of faults. IEC 60599-2015 09 [5] and Institute of Electrical and Electronics Engineers IEEE C57.104-2019 [6] standards provide several DGA interpretation methods achieved on the calculation of certain oil-dissolved flammable gas ratios. Between the methods proposed by these standards and the evolution of the effective faults in transformers, there is a correlation validated by large datasets collected from operating transformers and compared with those collected from out-of-service transformers. We briefly present the most common DGA interpretation methods proposed in [5] (IEC ratio and Duval 1 triangle (DTr-1)) and in [6] (key gas analysis, Doernenburg ratio, and Rogers ratio).

### 2.1. Key Gas Method

To determine the faults in the transformer, the key gas method uses the combination of individual gases ($H_2$; $CH_4$; $C_2H_6$; $C_2H_4$; $C_2H_2$; CO; $CO_2$) and the total concentration of flammable gases (TDCG = $H_2$ + $CH_4$ + $C_2H_6$ + $C_2H_4$ + $C_2H_2$ + CO), and the result of the diagnosis is based on the determination of the relative maximum values of the key gases in relation to the rest of the gases dissolved in the transformer oil [6]. This method can only forecast the following general types of errors: partial discharges (PD) in oil, overheated oil, cellulose overheating, and arcing in oil. Researchers consider this method as very conservative because, according to it, a transformer can operate safely, even if this interpretation method shows an imminent risk, provided that the speed of gas generation is not increasing steadily [1–3]. Due to these things, this method is not widely used as a tool for the efficient interpretation of the transformer faults based on the DGA results.

### 2.2. Doernenburg Reporting Method

This method is one of the oldest methods used to identify the initial faults in transformers. To apply this method, the first condition to be fulfilled is that at least one of the key gas concentrations ($H_2$; $CH_4$; $C_2H_6$; $C_2H_4$; $C_2H_2$) exceeds twice the concentration limits (L1), as shown in Table 1, and that one of the other two gases exceeds the limit value L1. After this condition is fulfilled, the four typical gas ratios ($CH_4/H_2$; $C_2H_2/C_2H_4$; $C_2H_2/CH_4$; $C_2H_6/C_2H_2$) will be used to examine the types of faults as shown in Table 2 [1–3,6].

**Table 1.** Concentration limits L1 of dissolved gases.

| Dissolved Gases | $H_2$ | $CH_4$ | CO | $C_2H_2$ | $C_2H_4$ | $C_2H_6$ |
|---|---|---|---|---|---|---|
| Concentration limits L1 | 100 | 120 | 350 | 1 | 50 | 65 |

**Table 2.** Fault analysis based on Doernenburg ratio.

| Case of Error | $CH_4/H_2$ | $C_2H_2/C_2H_4$ | $C_2H_2/CH_4$ | $C_2H_6/C_2H_2$ |
|---|---|---|---|---|
| Thermal decomposition | >1 | <0.75 | <0.3 | >0.4 |
| Corona (Low intensity partial discharge) | <0.1 | Insignificant | <0.3 | >0.4 |
| Arcing (High-intensity partial discharge) | >0.1–<1 | <0.75 | >0.3 | <0.4 |

### 2.3. Rogers Ratio and IEC Ratio Methods

One of the two DGA interpretation techniques recommended in [5] is the method of IEC ratios using the same gas ratios ($C_2H_2/C_2H_4$; $CH_4/H_2$; $C_2H_4/C_2H_6$), as in the case of the Rogers ratio method. Tables 3 and 4 show the values of the ratios and the types of faults related to them for the methods of the Rogers and IEC ratios [4–6].

**Table 3.** Analysis based on Rogers ratios.

| Type of Fault | $C_2H_2/C_2H_4$ | $CH_4/H_2$ | $C_2H_4/C_2H_6$ |
|---|---|---|---|
| Normal unit | <0.1 | >0.1 to <1 | <1 |
| Partial discharge | <0.1 | <0.1 | <1 |
| Arcing | 0.1 to 3.0 | >0.1 to <1 | >3 |
| Low thermal temperature | <0.1 | >0.1 to <1 | >0.1 to <3 |
| Thermal: <700 °C | <0.1 | >1 | >0.1 to <3 |
| Thermal: >700 °C | <0.1 | >1 | >3 |

**Table 4.** Analysis based on IEC ratios.

| Case | Characteristic Fault | $C_2H_2/C_2H_4$ | $CH_4/H_2$ | $C_2H_4/C_2H_6$ |
|---|---|---|---|---|
| PD | Partial discharge | Insignificant | <0.1 | <0.2 |
| D1 | Discharge of low energy | >1 | 0.1–0.5 | >1 |
| D2 | Discharge of high energy | 0.6–2.5 | 0.1–1 | >2 |
| T1 | Thermal: T < 300 °C | Insignificant | >1 but insignificant | <1 |
| T2 | Thermal: 300 °C < T < 700 °C | <0.1 | >1 | 1–4 |
| T3 | Thermal: T > 700 °C | <0.2 | >1 | >4 |

In the IEC 60599-2015 09 [5] standard, it is recommended that, when the values of the ratios are not within the threshold ranges and do not match any fault, the graphical representation in two or three dimensions of the gas concentrations should be used, so the type of fault may be the area in the vicinity of the undiagnosed case. Moreover, some identified faults are not precisely for the overlapping fault areas of cases D1 and D2. Therefore, one of the serious drawbacks of these methods is that part of the gas ratio obtained is not included in the specific range of values and, thus, the fault diagnostic fails to be assessed.

In conclusion, ratio-based methods, such as Doernenburg, Rogers, and IEC can only be used if there is a substantial amount of gas used with the ratio, otherwise the methods lead to ratio values outside the specific range and it will not be possible for the type of fault to be identified [1–4]. Therefore, these methods can be used to identify faults, rather than detect them.

### 2.4. Duval Triangle Methods

The DTr-1, also presented in [5,6] was proposed by Michel Duval in the early 1970s and is widely used for the analysis of dissolved gases in mineral oil-filled transformers. This method is based on the values of gases $CH_4$, $C_2H_4$, and $C_2H_2$, which also correspond to the increasing level of gassing in transformers. The seven areas presented in the DTr-1 correspond to the faults, which may be found in the transformers in service, and are presented in Table 5 [7].

The DTr-1 method proved to be quite efficient in obtaining the main type of fault occurring in the mineral oil-filled transformers in service. When the DGA results are at the boundary between two fault areas, it is difficult to distinguish which of the two faults is the real one. It has also been found that some oils tend to unpredictably generate low-temperature gases (between 80 °C and up to 200 °C) [3,5,9,11], and, thus, may interfere with the correct identification of faults in transformers. The use of the DTr-1 method for the analysis of the dissolved gases in a transformer with normal insulation aging leads to an error, because the result of the DGA will show a fault in the said transformer.

**Table 5.** Faults identified using the DTr-1.

| Code | Fault or Stress |
|------|-----------------|
| PD | Corona-type partial discharges |
| D1 | Low-energy discharges |
| D2 | High-energy discharge |
| T1 | Thermal faults not exceeding temperature 300 °C |
| T2 | Thermal faults temperature at 300 °C to 700 °C |
| T3 | Thermal faults exceeding 700 °C |
| DT | Combinations of electrical and thermal faults |

To eliminate these uncertainties, Michel Duval developed triangles 4 (DTr-4) and triangles 5 (DTr-5), also called "low-temperature fault triangles" [7]. They should only be considered as an addition to the information for DTr-1 and should not be used for faults D1 and D2, which have been identified using the DTr-1. The DTr-4 presented in Figure 1 is used to obtain more information about the faults identified using the DTr-1 as low-temperature faults such as: PD, T1, or T2 and uses "low-energy gases" [6,7,12]: $H_2$, $CH_4$, and $C_2H_6$.

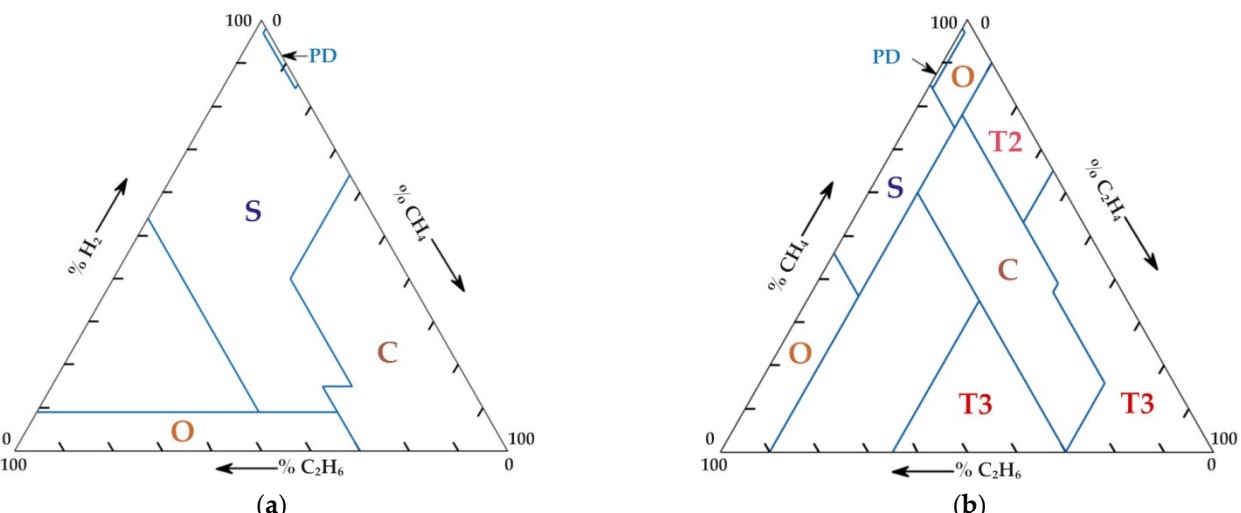

**Figure 1.** DTr-4 and DTr-5: (**a**) DTr-4 for low-temperature faults; (**b**) DTr-5 for thermal faults.

The defining of the fault areas in this triangle is shown in Table 6.

**Table 6.** Defining the fault areas in the DTr-5.

| Code | Fault or Stress |
|------|-----------------|
| PD | Corona-type partial discharges |
| S | Stray gassing of mineral oil |
| C | Paper carbonization caused by hot spots (T > 300 °C) |
| O | Overheating (T < 250 °C) |
| T2 | Thermal faults caused by temperature at 300 °C to 700 °C |
| T3 | Thermal faults occurring at very high temperatures (T > 700 °C) |

The DTr-5 for thermal faults is used to obtain more information about the faults identified using the Duval Triangle 1 as T2 or T3 type thermal faults and to confirm the faults accompanied by uncertainties after using the DTr-4.

This triangle uses "temperature gases" [6,7,12]: $C_2H_4$, $CH_4$, and $C_2H_6$ and is represented graphically in Figure 1. The defining of the fault areas in the DTr-5 is shown in Table 6.

### 2.5. The Duval Pentagons Method

A new method of interpreting the DGA is represented by the Duval Pentagons 1 and 2 (DPg-1 and DPg-2), using the five key gases ($H_2$; $CH_4$; $C_2H_6$; $C_2H_4$; $C_2H_2$) arranged at the apices of the pentagon. This method was created to solve problems which cannot be solved using the Duval Triangles method [3,6,8,9,12].

The DPg-1 and DPg-2 are shown in Figure 2, and the order of the gases at the apices of the pentagons corresponds to the increase in energy required for the generation of these gases ($H_2 \rightarrow C_2H_6 \rightarrow CH_4 \rightarrow C_2H_4 \rightarrow C_2H_2$) in the oil of a transformer in service, i.e., counterclockwise. This gas layout proved to be the most suitable in identifying the faults in transformers in the pentagon representation [10].

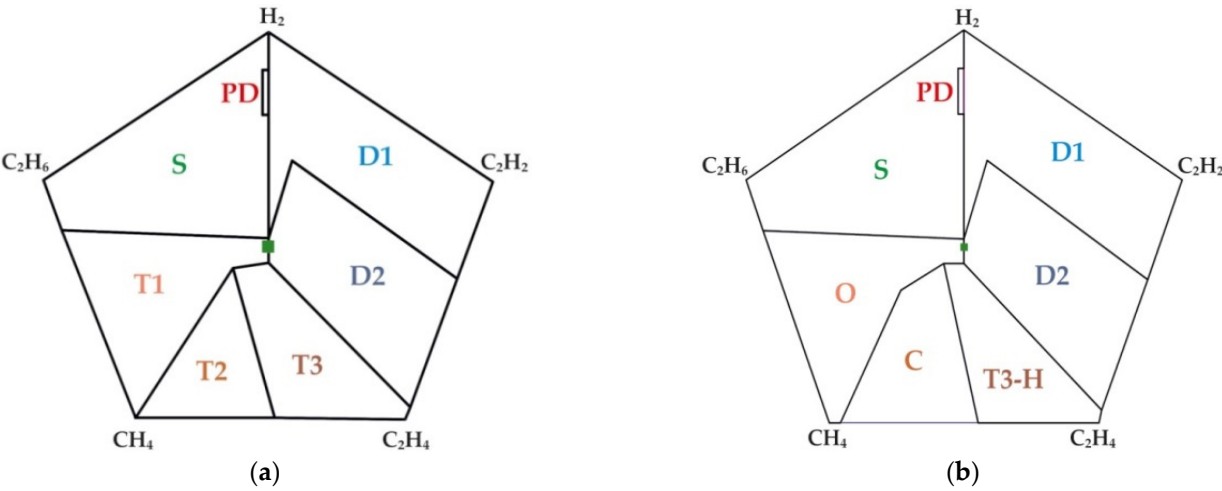

**Figure 2.** Representation of fault areas in the Duval Pentagon: (**a**) DPg-1; (**b**) DPg-2.

Figure 2a shows the DPg-1 with six main areas usually associated with the "basic" electrical or thermal faults used by the IEC standard, the IEEE standard and the DTr-1 (see Table 5) and area "S" of stray gas correlated with the generation of gases during the normal aging process of the complex insulation system of transformers.

When a thermal fault, such as T1, T2 or T3 occurs in DPg-1 after an analysis, it is important to know to what extent paper carbonization is involved in the occurrence of the fault, as this is an important factor in making an appropriate decision to avoid catastrophic damage [6,8,10]. So this problem can be solved by using the DPg-2, which is represented in Figure 2b, where the areas for the thermal faults are defined as follows:

- O: overheating < 250 °C;
- C: thermal faults with paper carbonization;
- T3-H: faults at high temperatures occurring only in oil.

It was found that the results of the DGAs, which are shown in the "C" zone of Pentagon 2 revealed the possible paper carbonization with a certainty of 100%, and, therefore, additional analyses with carbon oxides and furan compounds are needed for those transformers, to determine the level of degradation of the solid insulation [10].

When, for the same set of DGA results, DPg-1, DPg-2 and DTr-1, DTr-4, DTr-5 for mineral oils reveal different types of faults, this indicates that there is a combination of faults in the said transformer. Thus, DPg-1 and DPg-2 are not intended to replace DTr-1, DTr-4 and DTr-5, but to provide additional information that can help identify various faults, especially in the case of the combinations of faults [4,7,12].

The Combined Duval Pentagon method, obtained by overlapping DPg-1 and DPg-2 is described in [10]. The purpose of this combination is to make full use of both original pentagons and to make it easier for the faults in mineral oil-filled transformers to be automatically identified by using computer programs.

The proposed combined method presented in Figure 3 results in a number of 10 fault areas (areas S, PD, D1 and D2, which represent thermal problems are identical in both original pentagons) instead of 14 areas if both pentagons are used separately. The six areas with thermal problems presented in the Combined Duval Pentagon are defined in Table 7.

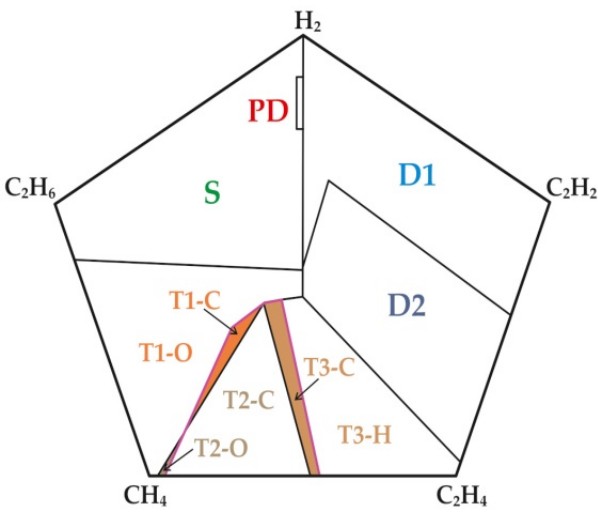

**Figure 3.** Duval Pentagon Combined with the ten fault zones.

**Table 7.** Defining the areas of thermal faults in the Combined Duval Pentagon.

| Code Combined Pentagon | Code Pentagon 1 | Code Pentagon 2 | Defining the Fault or Stress |
|---|---|---|---|
| T1-O | T1 | O | Confirmation of the thermal problem with the predicted temperature <300 °C, but with no carbonization of the solid insulation. |
| T1-C | T1 | C | Confirmation of the thermal problem with the predicted temperature less than 300 °C, but now with the probable paper implication, which shows the carbonization. |
| T2-O | T2 | O | Confirmation of the thermal problem, with a temperature between 300 °C and 700 °C, but unlikely to involve the solid insulation or the carbonization of the paper. |
| T2-C | T2 | C | Confirmation of the thermal problem with a temperature between 300 °C and 700 °C, with high probability of paper implication (probability of 80%, based on data from transformers with faults detected during the internal inspection). |
| T3-H | T3 | T3-H | Confirmation of the thermal emission only in oil, temperature range above 700 °C. |
| T3-C | T3 | C | Confirmation of the thermal problem at high temperatures (above 700 °C) with paper implication in the occurrence of the fault (carbonization). |

The Duval Combined Pentagon is not created to replace DPg-1 and DPg-2, but to supplement them and make easy the calculation of the 10 types of faults using software applications.

*2.6. Single Gas Ratio Method*

For the methods described above, References [5,6] have added three single gas ratios ($CO_2/CO$, $O_2/N_2$ and $C_2H_2/H_2$), which can be used as complementary methods for diagnosing transformer faults.

2.6.1. $CO_2/CO$ Ratio

In case of cellulose degradation, atmospheric air ingress or transformer oil oxidation, carbon monoxide (CO) and carbon dioxide ($CO_2$) are generated. Until recently, these carbon oxides were considered good indicators for the determination of paper involvement in the occurrence of a fault diagnosed in the transformer, but recent research conducted by specialists in the International Council on Large Electric Systems (CIGRE), IEC and IEEE working groups has shown that this is not always the case [5,6,12]. Therefore, the new viewpoints on the interpretation of carbon oxides and the $CO_2/CO$ ratio are:

- High values of CO (>1000 ppm) and/or ratios $CO_2/CO < 3$, with no significant amounts of other fault gases are not an indication of a C-type fault, i.e., paper carbonization, but it is due to the oxidation of the mineral oil under conditions that correspond to low oxygen content in oil;
- High values of CO (>1000 ppm) and ratios $CO_2/CO < 3$ in the presence of significant amounts of other fault gases and furans are considered as a confirmation of paper involvement in the occurrence of a fault, with possible carbonization;
- High values of $CO_2$ (>10,000 ppm) and ratios $CO_2/CO > 20$ and high values of furans (>5 ppm) show the slight overheating (<160 °C) with slow paper degradation until low values of the degree of polymerization (DP) of paper are reached. This does not prevent the transformer from operating normally, but there is concern that, due to the low DP of paper, the cellulose insulation will not withstand stresses, such as short circuits or transient overcurrent;
- In some cases, the faults located in small volumes of paper do not generate significant amounts of CO and $CO_2$ compared to the high amounts of the said gases in operation, but these faults generate significant amounts of other hydrocarbon gases, allowing the detection of faults in the paper using the Duval Pentagon 2 and the Duval Triangles 4 or 5.

Therefore, the implication of faults in the paper will not only be supported by CO and $CO_2$, but will also be verified by the generation of other gases or the analysis of furan compounds [4,10,12,13].

2.6.2. $O_2/N_2$ Ratio

Oxygen ($O_2$) and nitrogen ($N_2$) dissolved in oil are detected due to contact with atmospheric air in the tank of free-breathing transformers or air ingress through leakages in the sealed equipment. At equilibrium with air, the concentrations of $O_2$ and $N_2$ in oil are approximately 32,000 and respectively 64,000 ppm, and the $O_2/N_2$ ratio is approximately 0.5 [5,6].

The decrease in the oxygen concentration or the $O_2/N_2$ ratio during the functioning of the transformer shows the oxidation of the oil due to its overheating and, thus, this ratio can be used to confirm the thermal faults. An increase in the oxygen content or the $O_2/N_2$ ratio in the sealed transformers shows a leakage to the air conservation system (nitrogen blanket or membrane).

The normal value of the $O_2/N_2$ ratio is influenced by factors, such as: the type of transformer, the loading and conservation system used. Thus, the specialists of the CIGRE working group concluded that: the $O_2/N_2$ ratio is <0.2 for all of the nitrogen blanketed transformers, and approximately 60% of the membrane-sealed transformers, and $O_2/N_2$ ratio is >0.2 for all air breathing transformers, and the rest of the 40% of the membrane-sealed ones [12].

### 2.6.3. $C_2H_2/H_2$ Ratio

In the power transformers equipped with on load tap changers (OLTC), oil contamination may occur in the main tank, which leads to misdiagnosis, if there is a possibility of communication with the oil or gas in the OLTC tank and the transformer main tank or between the respective conservators.

Thus, the values between 2 and 3 of the $C_2H_2/H_2$ ratio are considered an indication of the contamination of the oil in the transformer main tank with the oil or gas from the OLTC [5,6]. In this case, the evaluation of the DGA results from the transformer main tank must be performed by decreasing the contamination in the OLTC or considered inconclusive [7,11,12].

Modern OLTCs are designed so that they can no longer lead to the contamination of the oil in the main tank of the power transformer.

### 2.7. C3 Hydrocarbon Method

The methods for the interpretation of the DGA presented above only consider C1 and C2 hydrocarbons. Some newer practical methods also use C3 hydrocarbon concentrations, and their authors consider them useful in making a more accurate diagnosis. C3 hydrocarbons are very soluble in oil and are not affected by the possible diffusion into the ambient air, and, because of this, the result of the analysis largely depends on the method of extraction used [4–6].

CIGRE specialists in [12] presented in detail methods for identifying faults using C3 hydrocarbons. Thus we can say that the additional ratios $C_3H_6/C_3H_8$ and $C_2H_4/C_3H_8$ are used to confirm the temperature range for the thermal faults as is presented in Table 8.

**Table 8.** Confirmation of the temperature range for the thermal faults.

| Gas Ratios | Temperature Range | | |
|---|---|---|---|
| | 150–300 | 300–700 | >700 |
| $C_3H_6/C_3H_8$ | <2 | 2–6 | <6 |
| $C_2H_4/C_3H_8$ | <3 | 3–15 | <15 |

### 2.8. Method of Furan Compounds

As mentioned above, to determine the faults in the power transformers, in view of establishing paper implication and the possible carbonization of the solid insulation, furans can also be used, in addition to carbon oxides, thus allowing the determination of the degree of degradation of the cellulose insulation [10,15,24–26].

Furans are a family of chemical compounds, which are detected as dissolved in the power transformer oil; they are not generated by it, but occur exclusively as a result of the degradation of the solid insulation and have an important significance in assessing the condition of the solid insulation of the power transformer in operation and implicitly in assessing its lifespan. Therefore, we can say that the furan compounds occur as a result of specific conditions, which develop inside the transformer, hence their occurrence and concentration may suggest a certain operating fault. The names of these furan compounds and the most common causes of their occurrence are presented in Table 9 [13].

**Table 9.** Possible causes of the specific presence of the furan compound.

| Furan Compound | Causes of Occurrence |
|---|---|
| 5-HMF (5-hydroxymethyl-2-furfuraldehyde) | Paper oxidation (aging and heating) |
| 2-FOL (2-furfuryl alcohol) | High paper moisture |
| 2-FAL (2-furfuraldehyde) | General overheating or normal aging |
| 2-ACF (2-acetyl furan) | Caused by lightning (rarely detected by tests) |
| 5-MEF (5-methyl-2-furfuraldehyde) | Severe local overheating (hot-spot) |

A detailed review of the state-of-the-art application in terms of the furan compounds, as well as the difficulties associated with the correlation between them and the actual DP of the paper, are presented in [13–15]. According to these considerations, we can conclude that 2-FAL (see Table 10) is the most important and most widely used furan compound in determining the DP of the solid insulation and, based on its concentration, the following interpretation was proposed for the condition of insulation depending on the DP [15,23–26]:

**Table 10.** Transformer condition according to the 2-FAL content.

| 2-FAL Content (ppm) | Degree of Polymerization (DP) of Paper |
| --- | --- |
| 0–0.1 | 1200–700 |
| 0.1–1 | 700–450 |
| 1–10 | 450–250 |
| >10 | <250 |

## 3. Description of the Intelligent System for Determining the Faults of Power Transformers Based on Neural Networks

The general block diagram of the intelligent system for determining the faults of power transformers based on neural networks (ISDFPT-NN) by using Deep Learning Toolbox from MATLAB is presented in Figure 4. The first step in detecting the faults of power transformers is to take oil samples which are then processed in the laboratory by gas chromatography and high-performance liquid chromatography (HPLC). Following the analysis of the samples from the laboratory, the concentrations of dissolved gases and furan compounds will be obtained. The second step is to enter the obtained concentrations in the ISDFPT-NN to get the types of faults.

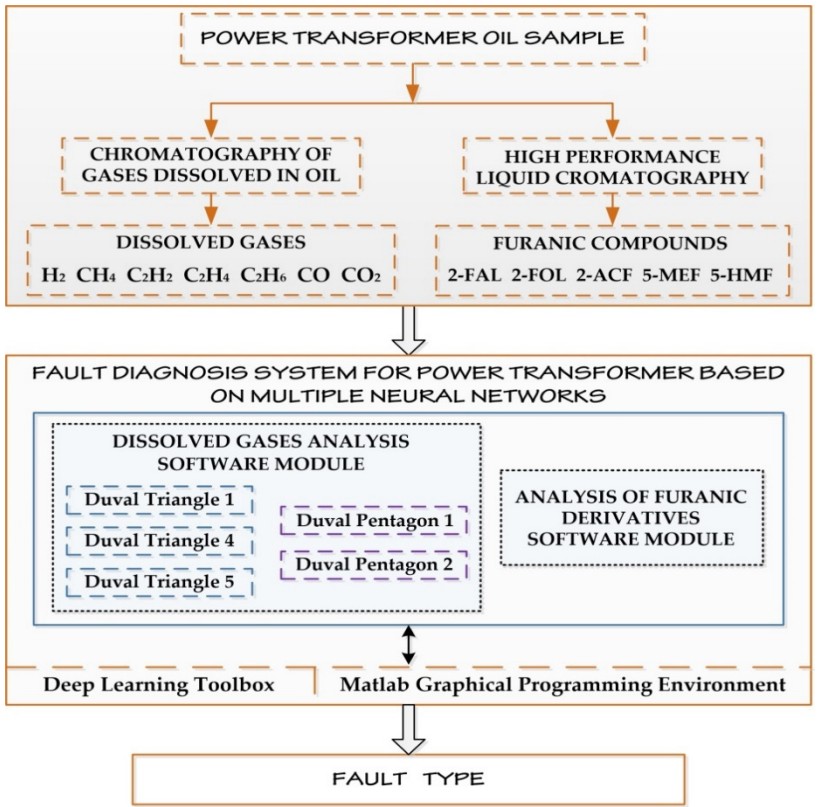

**Figure 4.** General scheme of the intelligent system for determining the faults of power transformers based on neural networks (ISDFPT-NN).

The ISDFPT-NN implements the flow chart of the diagnosis according to the methods described in Section 2 and is presented in Figure 5. RBFNNs were used for concentrations of $H_2$, $CH_4$, $C_2H_6$, $C_2H_4$ and $C_2H_2$, to implement the triangles and pentagons of the Duval method, and FFNNs were used for CO, $CO_2$ and furan compounds to implement the $CO_2/CO$ ratio method and the determination of DP, as is presented in Figure 5.

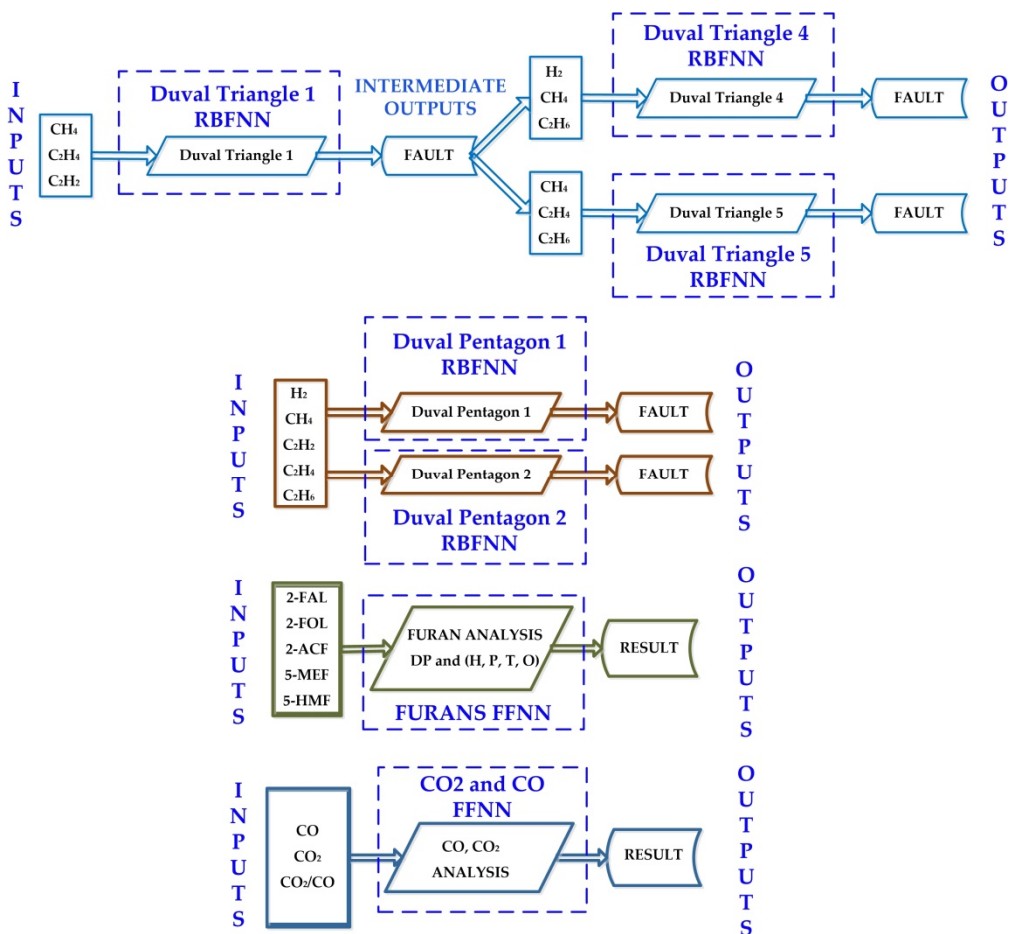

**Figure 5.** General flow chart of the ISDFPT-NN.

The 210 data samples were used in the implementation of ISDFPT-NN, which were taken from 94 power transformer units (step-up transformer, step-down transformer, distribution transformer, auto-transformers) with operating life between 20 and 35 years, from the laboratory database. These data samples were used to create, train, validate, and test the RBFNNs and FFNNs.

Based on the good results that can be achieved with the help of RBFNN with classification issues [27–29], for each method of detecting faults by means of triangles and pentagons, the Duval method has properly trained an RBFNN.

An RBFNN consists of three layers: the input layer (sensory layer), the hidden layer consisting of radial functions, which constitutes a basis for the input vectors, and the output layer. The transformation of the input space into intermediate layers is nonlinear and the transformation of the intermediate layer into the output layer is linear. A justification for these transformations is given by Cover's theorem on the classification of patterns [29]. Among the usual methods for training the RBFNN, the method of fixed and random centers is also chosen. The general form of a radial function that represents the output of a neuron from the hidden layer is expressed in the following form:

$$G\left(\|x - t\|^2\right) = \exp\left(-\frac{M}{d^2}\|x - t_i\|^2\right), i = 1, 2, \ldots, M \tag{1}$$

where *M*—number of centers, *d*–maximum distance between the centers, $X = (x_1, x_2, \ldots, x_n)^{\mathrm{T}}$, $x_i$ inputs, $\Omega = (t_1, t_2, \ldots, t_n)^{\mathrm{T}}$, $t_i$ input-associated weights.

Based on these, the neurons exit from the output layer based on its linearity in the following form:

$$Y = F(X) = \sum_{k=1}^{N} w_k G_k(\|X - \Omega\|) \tag{2}$$

where $w_k$—output-associated weights.

For example, the code for the training and creation in MATLAB of a corresponding RBFNN Duval Triangle 1 and the network structure and number of associated neurons are presented in Figure 6.

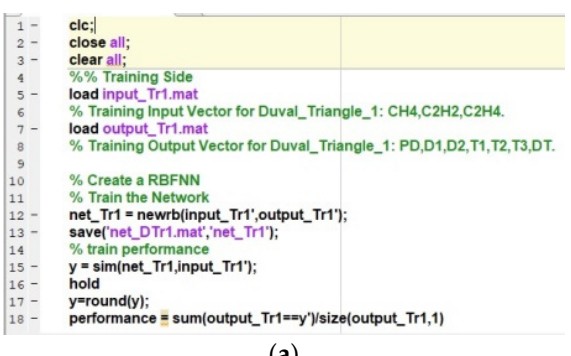

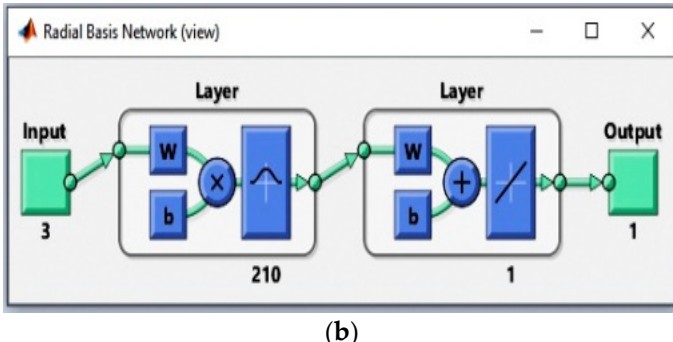

(a)                                          (b)

**Figure 6.** Example of the implementation of Duval_Triangle_1_RBFNN: (**a**) MATLAB code for creating and training the Duval_Triangle_1_RBFNN; (**b**) structure example of Duval_Triangle_1_RBFNN.

Similarly, the implementation of the RBFNNs corresponding to the methods presented in Sections 2.4, 2.6 and 2.8 was done as above, and the results of the training performances are presented in Figures 7–11. The performance represented is Mean Square Error (MSE) and is presented in Table 11.

$$MSE = \left( \frac{1}{N} \sum_{i=1}^{N} (A_i - T_i)^2 \right)^{1/2} \tag{3}$$

where *N*—number of observations, $T_i$—observed values, $A_i$—predicted values.

**Table 11.** Performances of the implemented neural networks.

| MSE | Value |
|---|---|
| Duval Triangle 1 | $4.4987 \times 10^{-29}$ |
| Duval Triangle 4 | $0.0248083$ |
| Duval Triangle 5 | $2.05548 \times 10^{-29}$ |
| Duval Pentagon 1 | $0.0119048$ |
| Duval Pentagon 2 | $0.01$ |

In case of training of each RBFNN, for the triangles and pentagons Duval method, the indicator of a training with good results is given by the parameter "Goal", which is given by the MSE defined by relation (3). In Figures 7–11 the parameter "Goal" is equal to zero, which means a very good training of the RBFNNs.

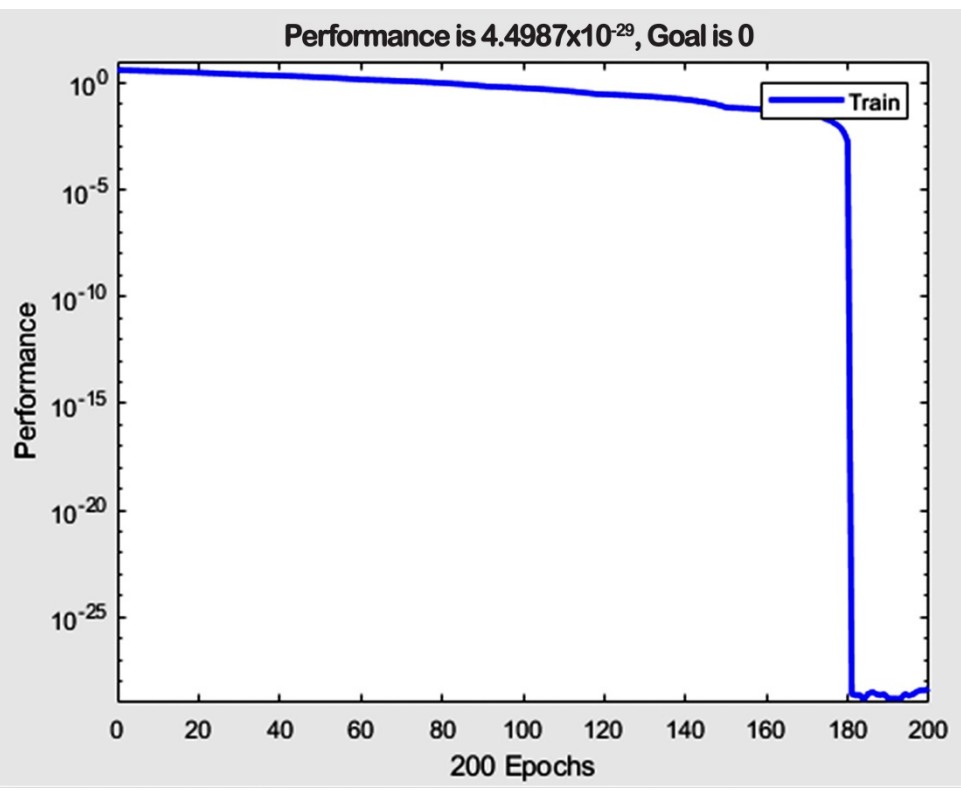

**Figure 7.** Performance of training for Duval_Triangle_1_RBFNN.

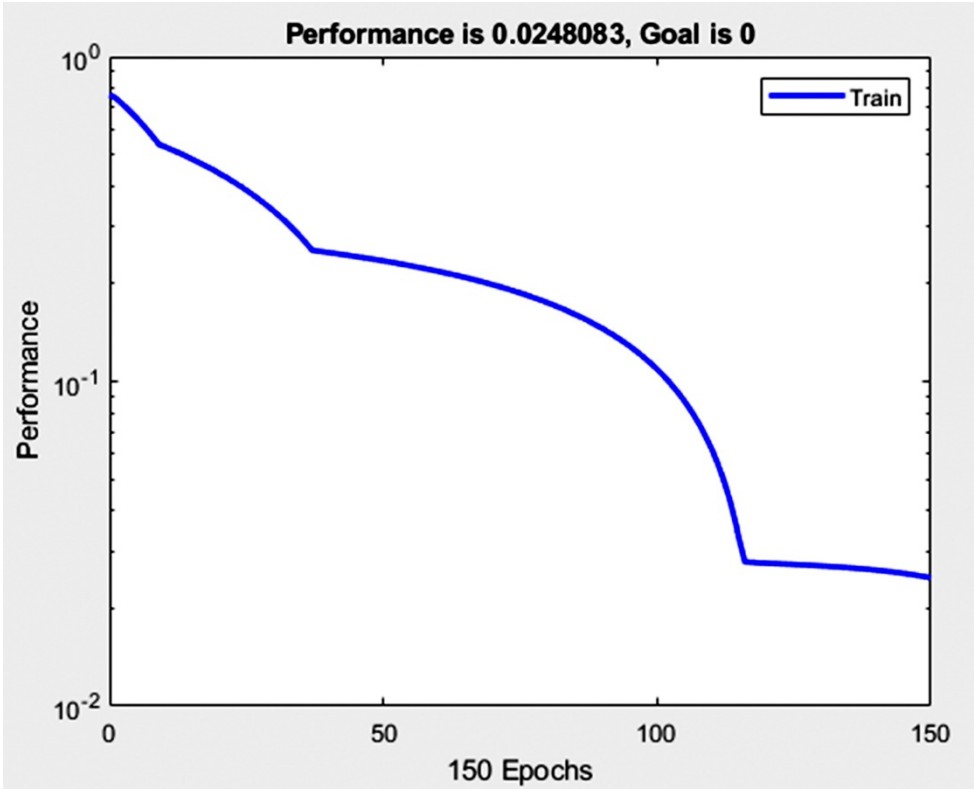

**Figure 8.** Performance of training for Duval_Triangle_4_RBFNN.

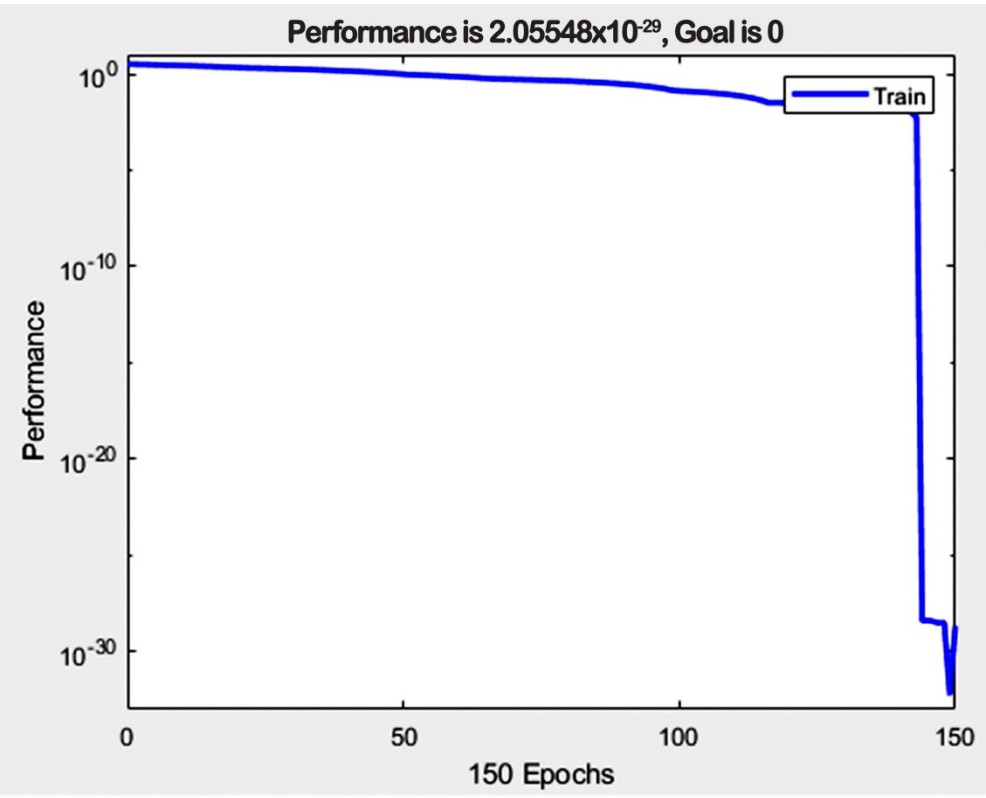

**Figure 9.** Performance of training for Duval_Triangle_5_RBFNN.

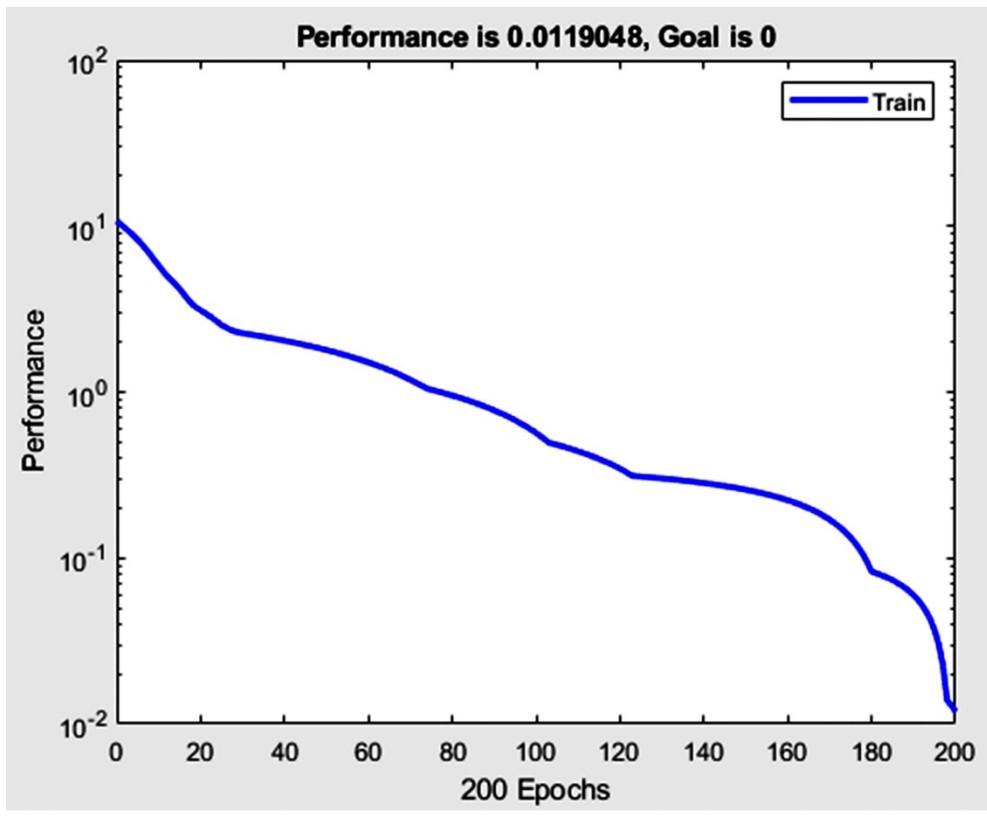

**Figure 10.** Performance of training for Duval_Pentagon_1_RBFNN.

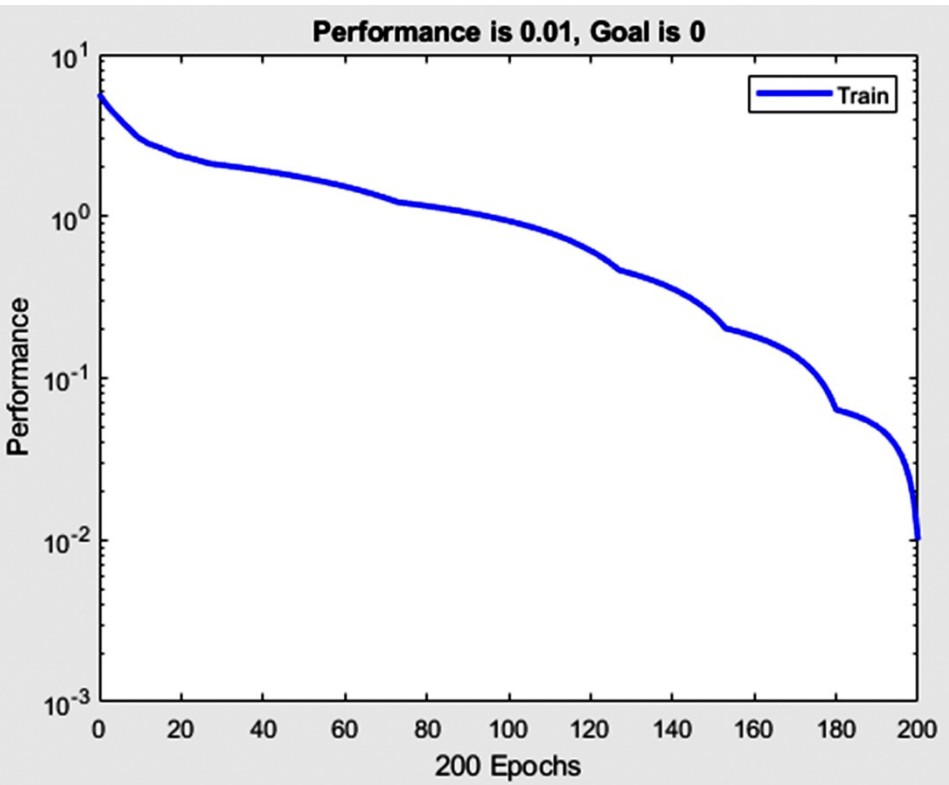

**Figure 11.** Performance of training for Duval_Pentagon_2_RBFNN.

The FFNN was used to determine cellulose degradation based on CO and $CO_2$ values. The special interest in this type of neural networks is due to their ability to operate with data different from those presented in the training stage and to learn from a random distribution of the synaptic weights of the network. The algorithm chosen for the neural network training is Levenberg–Marquardt, which is a method with a rapid convergence of the network, and is recommended for not very high input and output datasets [27,28,30].

For implementation, the Neural Network Fitting toolbox from the MATLAB programming environment is used [27,28]. CO and $CO_2$ input data and transformer state output data in terms of cellulose degradation are used as follows: 70% are used for training, 15% for validation, and 15% for testing. The neural network consists of two layers, one that represents the input layer and the other the output layer. For the input layer, 210 neurons similar to the RBNN type networks implemented for triangles and pentagons were set. The network's performances are shown in Figures 12 and 13. Regression factor R has a value close to 1, which denotes good training.

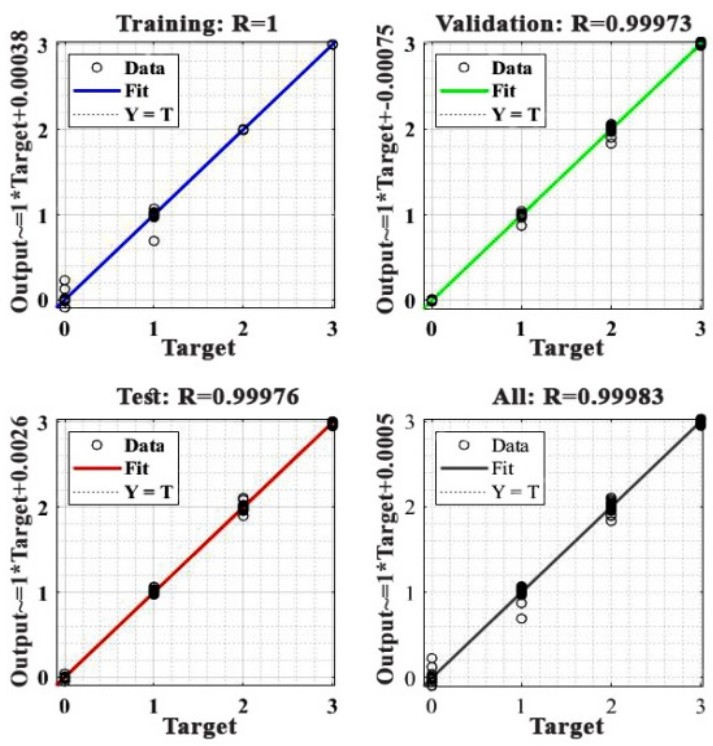

**Figure 12.** Regression parameter R for CO and $CO_2$ Analysis Feed Forward Neural Network (FFNN).

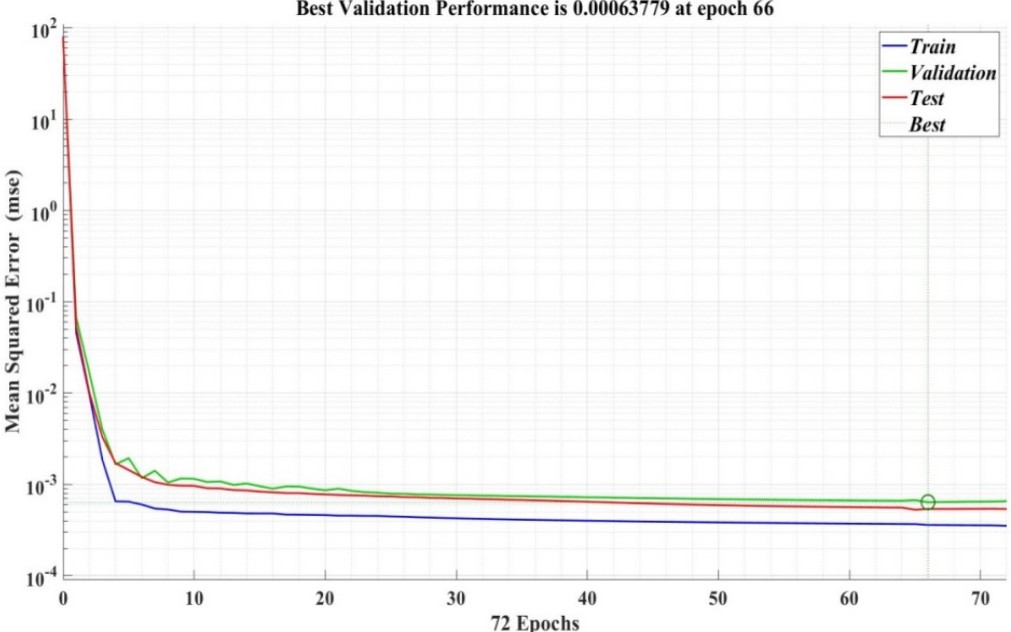

**Figure 13.** Performances of training for CO and $CO_2$ Analysis FFNN.

Similarly, the same type of neural network was chosen for the interpretation of the insulation state according to the degree of polymerization. The 2-FAL input and the transformer state output data in terms of the polymerization degree are used as follows: 70% are used for training, 15% for validation, and 15% for testing. The network's performances are presented in Figures 14 and 15.

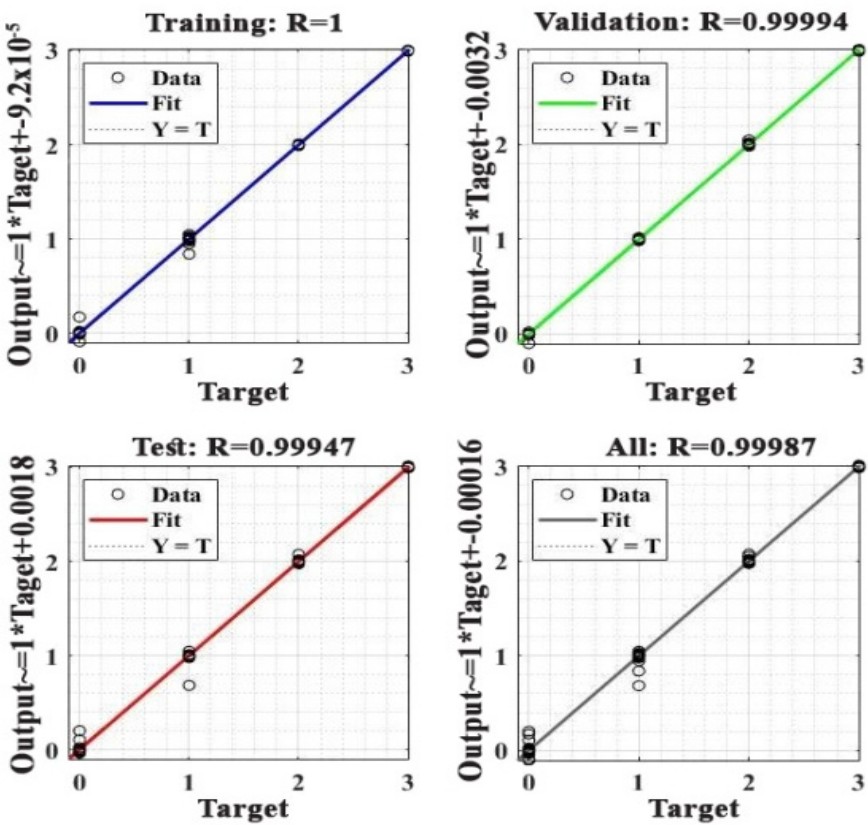

**Figure 14.** Regression parameter R for 2-FAL Analysis FFNN.

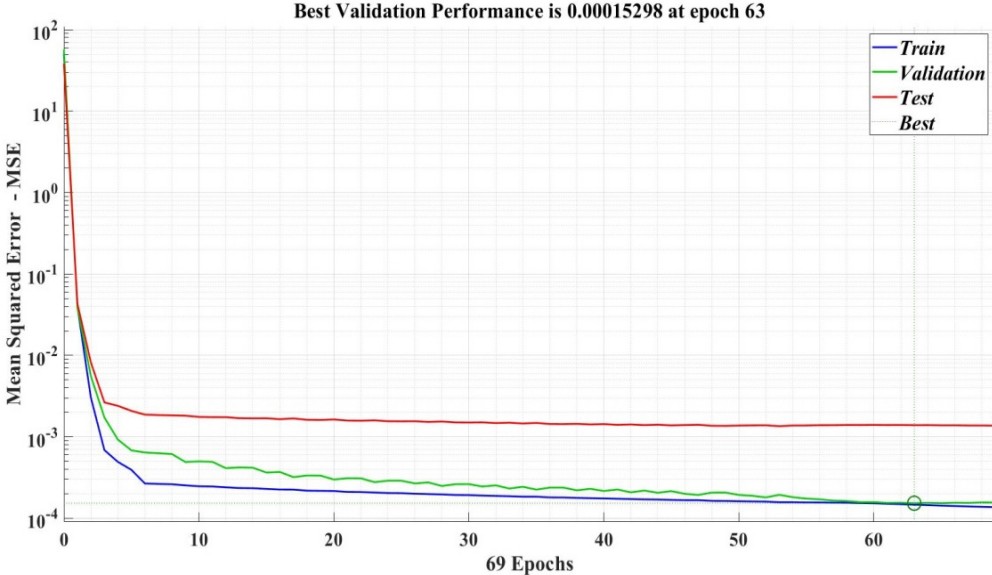

**Figure 15.** Performances of training for 2-FAL Analysis FFNN.

The block diagram of the implementation in MATLAB/Simulink of the main software module of ISDFPT-NN is shown in Figure 16. Each RBFNN and FFNN was created, trained, tested, and validated to implement the general flowchart in Figure 5.

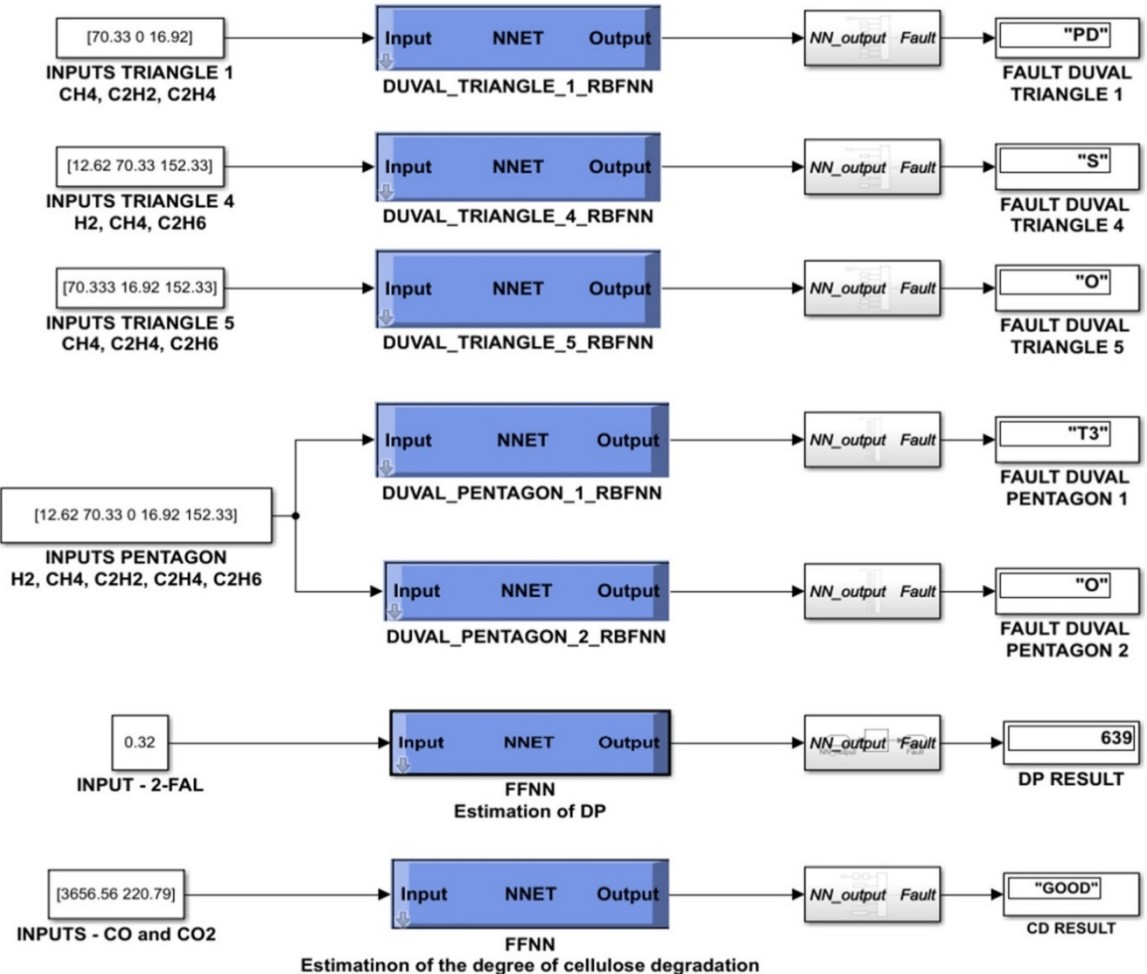

**Figure 16.** Neural software module of the ISDFPT-NN.

RBFNNs capable of classifying the faults of power transformers according to the dissolved gases present in the oil, following the triangles and pentagons Duval method were created, tested, and validated. Furthermore, using the Neural Network Fitting toolbox from MATLAB, FFNNs capable of classifying the faults of power transformers depending on CO and $CO_2$ gases and 2-FAL, following $CO_2/CO$ ratio and furan compounds methods were created, tested and validated. The software objects represented by RBFNNs and FFNNs created and trained are inserted in a Simulink subsystem (see Figure 16), which constitutes the software interface of ISDFPT-NN. This section presents an improvement of the classification algorithms, but, as in [24–26], ISDFPT-NN writes in a MySQL database in order to maintain a history, but also the usual communication facilities on the Intranet/Internet network.

## 4. Testing and Validation of the ISDFPT-NN

The validation of the proposed system was also done by comparing the results obtained against actual cases where certain refurbishments and repairs were performed.

Case 1. Free-breathing, mineral oil-filled three-phase step-down transformer, of 25 MVA rated power, 35/6.3 kV voltage ratio, and 32 years in service. The routine physical and chemical analyses, which also include the DGA (performed according to IEC 60599) and furan analysis showed that the oil is wet, strongly degraded, and the transformer has temperature points >700 °C (case of fault T3). The analysis using the proposed system revealed the faults presented in Tables 12 and 13, and the ratio $CO_2/CO < 3$ shows the degradation of the paper insulation.

**Table 12.** Dissolved Gas Analysis (DGA) analysis and fault identification.

| Dissolved Gas Concentration | | | | | | | Faults Identified | | | |
|---|---|---|---|---|---|---|---|---|---|---|
| $H_2$ | $CH_4$ | $C_2H_2$ | $C_2H_4$ | $C_2H_6$ | CO | $CO_2$ | Tr1 | TR5 | Pg1 | Pg2 |
| 27 | 70 | 10 | 306 | 81 | 1055 | 2849 | T3 | C | T3 | T3-H |

**Table 13.** Analysis of furans and their interpretation.

| Furans | ppm | Interpretation |
|---|---|---|
| 5-HMF | 0.345 | |
| 2-FOL | 0.0417 | |
| 2-FAL | 1.231 | Paper oxidation and local overheating. |
| 2-ACF | 0.0232 | |
| 5-MEF | 0.391 | |

The faults were confirmed after un-tanking, as follows: the phase A low voltage (LV) insulator had a loose nut, and the terminal was smutty with traces of overheating. Moreover, because of the loosening of the tightening nuts, the insulation of the hoses was damaged due to the local overheating of both the hoses and the terminal. This was consistent with the faults identified in Table 12 as a mixture of faults C and T3-H.

The faults were remedied by cleaning the terminal and the nut, by completely removing the degraded insulation and replacing it with new cotton tape, and finally, the tightening was done to the right torque.

Case 2. The routine physical and chemical analyses for the three-phase transformer in oil under load-oil natural air forced (TTUS-ONAF) step-down transformer, of 40 MVA rated power, 110/6.3 kV voltage ratio, and 27 years in service showed a wet oil, and the DGA (performed according to IEC 60599) showed that the transformer has high-energy electric arc discharges followed by: oil breakdown by arcing between coils or between terminals and earth or arcing in the OLTC along the contacts during switching, followed by oil leaks in the main tank. To determine whether or not the oil was contaminated by any leaks in the OLTC compartment, the $C_2H_2/H_2$ ratio was calculated, and its value of 2.3, according to IEC 60599, clearly shows the contamination.

The results obtained with the proposed system are presented in Tables 14 and 15.

**Table 14.** DGA analysis and fault identification.

| Dissolved Gas Concentration | | | | | | | Faults Identified | | |
|---|---|---|---|---|---|---|---|---|---|
| $H_2$ | $CH_4$ | $C_2H_2$ | $C_2H_4$ | $C_2H_6$ | CO | $CO_2$ | Tr1 | Pg1 | Pg2 |
| 51 | 83 | 118 | 73 | 39 | 534 | 4229 | D2 | D2 | D2 |

**Table 15.** Analysis of furans and their interpretation.

| Furans | ppm | Interpretation |
|---|---|---|
| 5-HMF | 0.345 | |
| 2-FOL | 0.072 | |
| 2-FAL | 0.71 | Paper oxidation. |
| 2-ACF | - | |
| 5-MEF | 0.0571 | |

It was decided to untank the transformer and the following was noted: the connections to the selector switch of the OLTC were loosened, and the hose insulation was smutty because of the local overheating caused by the electric arc formed during switching and, indeed, the tank of the OLTC was leaking.

The tank leak was repaired, and the selector switch of the OLTC was cleaned and the tightening of the connections on its plots was redone. The transformer conservator was

also replaced with an atmoseal conservator with two chambers, one for the transformer tank, and the other for the tank of the OLTC, each equipped with silica gel filters.

Case 3. For the step-up transformer of 63 MVA rated power, 10.5/121 kV voltage ratio, and 25 years in service, the routine measurements showed that the DGA analysis, according to the IEC 60599 standard indicates the thermal fault with temperatures of 150–300 °C (fault case T1). The analysis using the proposed system revealed the faults presented in Tables 16 and 17, and the ratio $CO_2/CO > 3$ and the furan analysis show a moderate degradation of the paper insulation.

**Table 16.** DGA analysis and fault identification.

| Dissolved Gas Concentration | | | | | | | Faults Identified | | | | |
|---|---|---|---|---|---|---|---|---|---|---|---|
| $H_2$ | $CH_4$ | $C_2H_2$ | $C_2H_4$ | $C_2H_6$ | CO | $CO_2$ | Tr1 | Tr4 | TR5 | Pg1 | Pg2 |
| 12.62 | 70.33 | 0 | 16.92 | 152.33 | 220.79 | 3656.56 | T1 | O | O | T1 | O |

**Table 17.** Analysis of furans and their interpretation.

| Furans | ppm | Interpretation |
|---|---|---|
| 5-HMF | 0.05 | |
| 2-FOL | 0.021 | |
| 2-FAL | 0.32 | Paper oxidation. |
| 2-ACF | - | |
| 5-MEF | - | |

This case is presented as the results of using the neural software module of the ISDFPT-NN, which are shown in Figure 16.

Because it was intended to make some improvements to the transformer, namely: mounting a Qualitrol-type pressure relief valve and replacing the free-breathing conservator with an atmoseal bag conservator, equipped with a separate compartment for the oil of the on-load tap-changer, the transformer was untanked. After the untanking, it was found that the insulation of the transformer is not smutty and shows no traces of degradation or carbonization, so it has a color specific to moderate aging; thus, confirming that the faults detected using the proposed system correspond to reality.

The ISDFPT-NN was tested on more than 80 power transformer units and compared to the results obtained in [24–26]. It was noted that there is an increase in accuracy from 95% to 95.7% and precision from 93% to 93.5%.

## 5. Conclusions

Since power transformers are key pieces of equipment in the electricity transmission and distribution systems, it is very important to diagnose their operating conditions and identify as accurately and early as possible the transformer failures. Ratio-based methods, such as Doernenburg, Rogers, and IEC can only be used if there is a substantial amount of gas used in the ratios, otherwise these methods lead to ratio values outside of the specific range, and the type of malfunction cannot be identified. Therefore, these methods can be used to identify faults rather than detect them. The methods of Duval triangles and pentagons can be used with a high percentage of accurate predictions compared to classical known methods (key gases, IEC reports, Rogers, Doernenburg), because, in addition to the six types of basic faults, they also identify four subtypes of thermal faults that provide complementary information, which is very important for the appropriate corrective actions to be applied to the transformer.

In this article, a new approach was proposed consisting of the complementarity between the analysis of dissolved gases in the oil of power transformers and the analysis of furan compounds, in order to identify the operating conditions of the power transformers, according to the paper degradation condition using artificial neural networks of the RBFN

and FFNN type. Moreover, 210 data samples were used in the implementation of the ISDFPT-NN, which were taken from 94 power transformer units (step-up transformer, step-down transformer, distribution transformer, auto-transformers) with operating lives between 20 and 35 years, from the laboratory database, and have been tested on more than 80 power transformer units, and compared to the results obtained in previous researches. It has been noted that there is an increase in accuracy, from 95% to 95.7%, and precision from 93% to 93.5%.

Proposals for future work consist of the development of a diagnostic system that also includes: the influence of oxygen, nitrogen, and their ratios, methods of interpreting faults using C3 hydrocarbon gases, and interfering with methods of diagnosing the conditions of the adjustment switch under loads, since it was found that transformer faults could occur mainly in = transformer insulation systems, transformer windings, bushings, and the OLTC.

**Author Contributions:** Conceptualization, A.-M.A., C.-I.N., M.N. and M.-C.N.; data curation, C.-I.N., M.N. and M.-C.N.; formal analysis, A.-M.A., C.-I.N., M.N. and M.-C.N.; funding acquisition, M.N.; investigation, A.-M.A., C.-I.N., M.N. and M.-C.N.; methodology, A.-M.A., C.-I.N., M.N. and M.-C.N.; project administration, M.N.; resources, A.-M.A. and M.N.; software, C.-I.N. and M.N.; supervision, A.-M.A., C.-I.N., M.N. and M.-C.N.; validation, A.-M.A., C.-I.N., M.N. and M.-C.N.; visualization, A.-M.A., C.-I.N., M.N. and M.-C.N.; writing–original draft, A.-M.A., C.-I.N., M.N. and M.-C.N.; writing—review and editing, C.-I.N., M.N. and M.-C.N. All authors have read and agreed to the published version of the manuscript.

**Funding:** The paper was developed with funds from the Ministry of Education and Scientific Research-Romania as part of the NUCLEU Program: PN 19 38 01 03 and grant POCU380/6/13/123990, co-financed by the European Social Fund within the Sectorial Operational Program Human Capital 2014–2020.

**Institutional Review Board Statement:** Not applicable.

**Informed Consent Statement:** Not applicable.

**Data Availability Statement:** Data sharing is not applicable.

**Conflicts of Interest:** The authors declare no conflict of interest.

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
