# Peer review of "Complementary Analysis for DGA Based on Duval Methods and Furan Compounds Using Artificial Neural Networks"

_energies, doi:10.3390/en14030588_

Round 1

Reviewer 1 Report

The article presents the application of ANN algorithm based on well-known and proven methods of transformer condition assessment based on oil diagnostics. To improve the judgment, the proposed algorithm utilizes the CO2/CO ratio assessment, the analysis of furan compound content and the state-of-the-art techniques for DGA evaluation – Duval Pentagons method.

The validation results for the algorithm created by Authors show its practical use. Additionally, the 3 case studies were presented for transformers that underwent internal inspection to further validate the findings.

This paper contains several minor flaws or points of concern such as:

  • In point 2.7 (C3 hydrocarbons) the ratios presented in table 8 (row 311) are not in line with the ratios mentioned before (row 309).
  • Point 3 – lack of information on the operation of the transformers used in this experiment (for example – distribution, generator step-up, etc.).
  • Point 3 – no information on number of transformers that were subjected to ANN training, testing & validation, only the percentage ratios are shown.
  • Point 4 – lack of some specific information (such as age, operation type) on the transformers included in the case studies.
  • Point 4 – low amount of algorithm output. Is it due to small amount of data from the units that underwent internal inspection?
  • The paper requires minor language correction (for example – row 479, passing insulators -> bushings, etc.).

To summarize, the article is substantial and is suitable for publication. However, it is recommended to correct the abovementioned shortcomings in order to further improve it.

Author Response

Dear reviewer, thanks for your recommendations.

  • In point 2.7 (C3 hydrocarbons): we have corrected the ratios presented in Table 8 and are in line with the ratios mentioned before.
  • Point 3: we added information and number of the transformers used in this experiment: “In the implementation of ISDFPT-NN were used 210 data samples, taken from 94 power transformer units (step-up transformer, step-down transformer, distribution transformer, auto-transformers) with operation age between 20 and 35 years, from laboratory database.”
  • Point 4: we added specific information on the transformers included in the case studies: “Case 1: Free-breathing, mineral oil-filled three-phase step-down transformer, with 25 MVA rated power, 35/6.3 kV voltage ratio, and 32 age in operation.”; “Case 2. The routine physical and chemical analyses for the TTUS-ONAF step-down transformer, with 40MVA rated power, 110/6.3 kV voltage ratio, and 27 age in operation showed a wet oil, and the DGA…”; “Case 3. For the step-up transformer with 63 MVA rated power, 10.5/121 kV voltage ratio, and 25 age in operation the routine measurements showed that the DGA analysis…”
  • Point 4: we added a clarification about the algorithm: “This section presents an improvement of the classification algorithms, but as in [24-26] ISDFPT-NN writes in a MySQL database in order to maintain a history, but also the usual communication facilities on the Intranet/Internet network.”

We corrected the expression “passing insulators” “bushings”.

Reviewer 2 Report

Overall, very interesting paper and approach!  Here are some comments:

Line 19:  "analyzes" should be "analyses"

Line 28:  "neuronal" to "neural"

Line 57:  "DET" not defined

Line 75:  "modified differential evolution whale optimization algorithm (MDE-WOA) needs a reference

Line 83:  What do you mean by different difficulties?

Line 149:  You have a divide sign in Table 2 bottom entry next to (high-intensity partial discharge)

Line 157:  Divide sign also used in this table 

Line 257:  What is CIGRE?

Line 311:  This also has a divide sign.  Are you using this for a range?  If so, a dash is more appropriate.

Line 337: Remove the dash in "g-eneral"

Line 341:  These figures need to be discussed and explained.  There should not be "back-to-back" figures without explanations and there needs to be a transition paragraph after the last figure.

Line 349:  Remove "a" from "a linear."

Line 356:  "...exits..." to "...exit..."

Line 357:  "k" should be a subscript

Line 365:  Remove "the" in front of "the figures"

Line 366:  Remove "the" in front of "the table"

Lines 369 - 375:  These figures need more explanation.  For instance why is the performance for Figures 8, 10, and 11 so different from Figures 7 and 9?  A little explanation under each figure would be nice.

Line 376:  What is FFFN?

Lines 389 and 390:  These figures require further explanation and discussion underneath each figure.

Lines 395 - 396:  These figures require further explanation and discussion underneath each figure.

Line 399:  This figure requires more explanation.  It is nice but why does the reader need to look at it?   What should we get from this figure?

Line 405:  "analyzes" to "analyses"

Line 420:  "analyzes" to "analyses"

Line 425:  What is OLTC?

Line 442:  Divide sign should be dash?

Line 445:  Sentence has incorrect grammar.

Line 458:  How much of an increase in accuracy and precision?

Line 463:  "...substantial amount of gas used in the report(?)..."

Author Response

            Dear reviewer, thanks for your recommendations.

            We made the corrections of the grammatical mistakes.

            CIGRE - International Council on Large Electric Systems.

            OLTC - on load tap changers

            We added in text more explanations about Figures 7-16.

            Regarding the increasing in accuracy and precision of the proposed system we added: “The ISDFPT-NN has been tested on over 80 power transformers units and compared to the results obtained in [24-26] there is an increase in accuracy from 95% to 95.7% and precision from 93% to 93.5%.”

            Line 463:  "Ratios based methods such as Doerenburg, Roger and IEC can only be used if there is a substantial amount of gas used in the ratios, otherwise these methods lead to ratio values outside the specific range and the type of malfunction cannot be identified."

Reviewer 3 Report

See attached comments

Author Response

            Dear reviewer, thanks for your recommendations.

            We corrected the graph of combined pentagons.

            We add in references the paper “Duval, M.; De Pablo, A.; Atanasova-Hoehlein, I.; Grisaru, M. Significance and detection of very low degree of polymerization of paper in transformers. IEEE Electrical Insulation Magazine 2017, 33(1), pp. 31–38.” and we made the corrections.

            Following yours suggestion we corrected with “<3” and we added the proposed text.

            In Tables 13, 15, and 17 we made the corrections.

Reviewer 4 Report

The manuscript “Complementary Analysis for DGA Based on Duval Methods and Furan Compounds using Artificial Neural Networks” is poorly written. At the current state, it is difficult to understand the contents. Hence, it is difficult to assess the research contributions, even to follow the discussions. This reviewer recommends a complete rewrite and thorough review by the authors and/or an English editor before the manuscript can be submitted for a journal review process.

In addition, the following comments may be useful to improve the manuscript quality:

  1. Very serious writing errors include, among others, but not limited to:
  • Long sentences that cause confusion
  • Dangling modifiers
  • Separations between subject and predicate
  • Single sentence paragraphs
  • Wrong passive voices, for example: “In [16] is proposed…”
  1. Significant results, including quantitative results, should be included in the abstract and conclusions.
  2. Research contributions should be explicitly listed in bullet or enumerated points
  3. The reviewer is not clear about the original work from the authors. The abstract mentioned that the new approach is based on the complementary between the analysis of the gas dissolved in the transformer oil and the analysis of furan compounds. Did the author just combine the two existing analyses? Or is the focus on building the intelligent system for determining the faults as mentioned in Section 3? If that is so, the authors should have focused the writing on this work. The focus should also be mentioned in the abstract, introduction and, conclusions.
  4. More detail explanation should be given about how this work continues the previous works in [23-25].

Author Response

            Dear reviewer, thanks for your recommendations.

            The whole article was revised from the point of view of English grammar.

            Following yours suggestions we made the changes in text (see the track changes).

            This paper is a continuation of previous works and we improved the approach of methods and we implemented in ISDFPT-NN software system. Thus in this article was proposed a new approach consisting of the complementarity between the analysis of dissolved gases on power transformer oil and the analysis of furan compounds, in order to identify the state of operation of the power transformers according to the paper degradation state using artificial neural networks of the type RBFN and FFNN. Finally we obtain an increasing of the accuracy and precision.

Round 2

Reviewer 1 Report

I've not any remarks.

Author Response

Dear reviewer, thanks for your assessments.

Reviewer 4 Report

The authors have tried to improve the writing. However, it still needs significant improvement. The most obvious one is the repeated uses of single sentence paragraphs and long sentences. A paragraph must contains a main idea and at least one supporting sentence. Hence, it cannot be a single sentence. In addition, the long sentences are likely to cause confusion. This reviewer understands that the review focus should be on the technical work. However, a manuscript must be in good writing form before the technical contents can be reviewed well. 

This reviewer would also like to mention that the authors' response to the reviewer comments should have been more focused. The revisions in responses to the comments that can be pointed out in the revised manuscript should have been pointed out clearly. The authors also ignore few comments. For example, the author did not addressing any thing about the quantitative results in the abstract and conclusions.

Author Response

            Dear reviewer, thanks for your recommendations.

            The whole article was revised from the point of view of English grammar.

            Following yours suggestions we made the changes in text (see the track changes).

            This paper is a continuation of previous works and we improved the approach of methods and we implemented in ISDFPT-NN software system. The methods of Duval triangles and pentagons can be used with a high percentage of correct predictions compared to the known classical methods (key gases, IEC, Rogers, Dornerburg ratios), because, in addition to the six types of basic faults, they also identify four sub-types of thermal faults which provide very important additional information for the appropriate corrective actions to be applied to the transformer. A new approach is presented based on the complementarity between the analysis of the gases dissolved in the transformer oil and the analysis of furan compounds, for the identification of the different faults, especially when there are multiple faults, by extending the diagnosis of the operating condition of power transformers in terms of paper degradation. The implemented software system based on artificial neural networks has been tested and validated in practice with good results. 210 data samples were used in the implementation of the ISDFPT-NN, which were taken from 94 power transformer units (step-up transformer, step-down transformer, distribution transformer, auto-transformers) with operating life between 20 and 35 years, from the laboratory database, and has been tested on more than 80 power transformers units and compared to the results obtained in previous researches. It has been noted that there is an increase in accuracy from 95% to 95.7% and precision from 93% to 93.5%.
